# TOWARDS RELAXING THE UNBIASEDNESS CONDITION OF DOUBLY ROBUST ESTIMATORS FOR DEBIASED RECOMMENDATION

## ABSTRACT

Recommender system aims to recommend items or information that may be of interest to users based on their behaviors and preferences. However, there may be sampling selection bias in the process of data collection, i.e., the collected data is not a representative of the target population. Many debiasing methods are developed based on pseudo-labelings. Nevertheless, the effectiveness of these methods relies heavily on accurate pseudo-labelings (i.e., the imputed labels), which is difficult to satisfy in practice. In this paper, in contrast to the existing doubly robust estimators that take *strictly accurate* pseudo-labelings as an unbiasedness condition, we theoretically propose several novel doubly robust estimators that are unbiased when either (a) the pseudo-labelings *deviate from* the true labels with an arbitrary user-specific inductive bias, item-specific inductive bias, or a combination of both, or (b) the learned propensities are accurate. We further propose a principled propensity reconstruction learning approach that adaptively updates the constraint weights using an attention mechanism and effectively controls the variance. To summarize, the proposed methods greatly relax the unbiasedness condition of the widely-adopted doubly robust estimators, which empirically result in much lower bias. Extensive experiments show that our approach outperforms the state-of-the-art on one semi-synthetic dataset and three real-world datasets.

## 1 INTRODUCTION

By analyzing users' historical behaviors and preferences, recommender system (RS) predicts and recommends items or information that users may like. However, as users are free to choose which item to rate, so that the collected data is always not a representative of the target population (or inference space) (Schnabel et al., 2016; Wang et al., 2019; Chen et al., 2022; Wu et al., 2022; Saito and Nomura, 2022), and similar findings occur with tasks such as post-click conversion rate (pCVR) prediction (Guo et al., 2021; Dai et al., 2022), post-view click-through & conversion rate (CTCVR) prediction (Ma et al., 2018; Zhang et al., 2020; Wang et al., 2022), and uplift modeling (Saito et al., 2019; Sato et al., 2019; 2020). Since the labels are observable only in the collected data and missing in the target population, this poses a great challenge to achieve unbiased learning (Wang et al., 2020a).

To address this problem, the error-imputation-based (EIB) methods (Hernández-Lobato et al., 2014) first impute the missing labels and then train the prediction model using both the observed labels and pseudo-labelings. However, as the pseudo-labeling model is trained with observed data while deployed in the missing data, it is difficult to obtain accurate pseudo-labelings, leading to sub-optimal performance (Dai et al., 2022). The inverse-propensity-scoring (IPS) methods inversely weights the prediction error for each observed rating with the propensity of observing that rating (Schnabel et al., 2016), but it is empirically difficult to set proper propensity scores and theoretically has greater variance (Saito, 2020). By utilizing both pseudo-labeling and propensity models, the doubly robust (DR) method is proposed to weaken the unbiasedness condition of the EIB and IPS estimators (Wang et al., 2019), with many enhanced DR approaches developed (Guo et al., 2021; Zhang et al., 2020; Chen et al., 2021; Wang et al., 2021; Dai et al., 2022; Wang et al., 2022). The advantage of DR estimators is attributed to the property of double robustness, i.e., it is unbiased if either the learned propensities or the pseudo-labelings are accurate. We summarize the unbiasedness condition of the previous debiasing estimators in Table 1 (see Appendix A for more discussions on related works).

Table 1: Comparison of the various debiasing methods, where $\hat{p}$ and $\tilde{r}$ denotes the learned propensities and pseudo-labelings, respectively. The red and blue color highlight the unbiasedness of the proposed DR estimator under arbitrary user-specific inductive bias $b_u$ and item-specific inductive bias $b_i$.

| Method | Unbiasedness Condition |
|---|---|
| EIB | $\tilde{r}_{u,i} = r_{u,i}$ |
| IPS, Multi-IPS, ESCM$^2$-IPS | $\hat{p}_{u,i} = p_{u,i}$ |
| DR, Multi-DR, DR-JL, MRDR, ESCM$^2$-DR | $\hat{p}_{u,i} = p_{u,i}$ or $\tilde{r}_{u,i} = r_{u,i}$ |
| User-DR (ours) | $\tilde{p}_{u,i} = p_{u,i}$ or $\tilde{r}_{u,i} = r_{u,i} + b_u$, for all $b_u \in \mathbb{R}$ |
| Item-DR (ours) | $\tilde{p}_{u,i} = p_{u,i}$ or $\tilde{r}_{u,i} = r_{u,i} + b_i$, for all $b_i \in \mathbb{R}$ |
| User-Item-DR (ours) | $\tilde{p}_{u,i} = p_{u,i}$ or $\tilde{r}_{u,i} = r_{u,i} + b_u + b_i$, for all $b_u, b_i \in \mathbb{R}$ |

Note: $b_u$ and $b_i$ are arbitrary user-specific and item-specific inductive biases (see section 3 for more details). Since the proposed methods require to reconstruct the learned propensities, so we use $\tilde{p}_{u,i}$ instead of $\hat{p}_{u,i}$ for distinguish.

Despite the double robustness providing additional protection against inaccurate pseudo-labelings, recent studies have shown that DR methods are highly sensitive to inaccurate pseudo-labelings. Specifically, when the learned propensities are slightly inaccurate, the DR estimator can be severely biased with inaccurate pseudo-labelings (Kang and Schafer, 2007; Molenberghs et al., 2015; Vermeulen and Vansteelandt, 2015; Seaman and Vansteelandt, 2018). These inaccurate pseudo-labels would as a result lead to biased prediction models during the training phase (Wen et al., 2022; Mansoury et al., 2020; Krauth et al., 2022). Therefore, we believe it is essential to develop novel DR estimators with relaxed unbiasedness conditions on the accurate pseudo-labelings.

To this end, in this paper we theoretically propose several novel DR estimators that are *unbiased under inaccurate pseudo-labelings*, named User-DR, Item-DR, and User-Item-DR. As shown in Table 1, our theoretical analysis proves that the User-DR estimator is unbiased as long as the pseudo-labelings *deviate* the true labels with an arbitrary *user-specific inductive bias*. Whereas the corresponding unbiasedness condition of previous DR estimators require the pseudo-labelings to be *strictly equal* to the true labels, which is much stronger than that of the proposed User-DR estimator. Similarly, the Item-DR (or User-Item-DR) estimators are unbiased as long as the pseudo-labelings *deviate* the true labels with an arbitrary *item-specific inductive bias* (with an arbitrary *user-specific inductive bias*). In addition, similar to the DR estimators, all the User-DR, Item-DR, and User-Item-DR estimators are unbiased if the learned propensities are accurate. Table 1 compares the proposed DR estimators with the existing estimators, and it is clear that the newly proposed DR estimators greatly relaxes the conditions for achieving unbiasedness and notably alleviate the risk of inaccurate pseudo-labelings.

In fact, the requirements on pseudo-labelings for achieving unbiasedness in the proposed estimators are realistic and have important implications. In RS, the user-item interactions are influenced by various factors, such as user herding effect (Liu et al., 2016), user conformity (Zheng et al., 2021), item popularity (Zhang et al., 2021), and item exposure position (Ai et al., 2018). These user-specific and item-specific factors may cause user-specific and item-specific inductive biases in the pseudo-labeling model. Remarkably, the proposed DR estimators can still achieve unbiased learning, even if the user-specific and item-specific inductive biases are *arbitrary* and *unknown*. Based on this, we further propose a principled propensity reconstruction learning approach that alternatively updates the propensity model, the imputation model, and the prediction model, where the constraint weights in the propensity reconstruction loss are adaptively updated using an attention mechanism. To summarize, the proposed methods greatly relax the unbiasedness condition of the widely-adopted doubly robust estimators, which empirically result in much lower bias.

The main contributions of this paper are:

- We theoretically propose several novel DR estimators that are unbiased when the pseudo-labelings deviate from the true labels with an arbitrary and unknown user-specific inductive bias, item-specific inductive bias, or a combination of both.

- We further propose a principled propensity reconstruction learning approach that adaptively updates the constraint weights using an attention mechanism, and theoretically demonstrates the variance of our DR estimators are highly controllable and manageable.

- We perform semi-synthetic experiments to verify the effectiveness of the proposed methods for arbitrary user-specific and item-specific inductive bias, while previous methods fail to unbiasedly estimate the ideal loss. We also conducted extensive experiments on three real-world datasets to demonstrate the advantages of the proposed methods.

## 2 PRELIMINARIES

Let $\mathcal{U} = \{u_1, u_2, \ldots, u_m\}$ be the set of $m$ users, $\mathcal{I} = \{i_1, i_2, \ldots, i_n\}$ be the set of $n$ items, and $\mathcal{D} = \mathcal{U} \times \mathcal{I}$ be the set of all user-item pairs. Denote $\mathbf{R} \in \mathbb{R}^{m \times n}$ as the rating matrix of all user-item pairs, where $r_{u,i}$ indicates the rating of user $u$ on item $i$. Let $x_{u,i}$ be the feature of user $u$ and item $i$, and $\hat{\mathbf{R}} \in \mathbb{R}^{m \times n}$ be the rating prediction matrix for $\mathbf{R}$, where $\hat{r}_{u,i} = f(x_{u,i}; \theta)$ is the predicted rating induced by a prediction model, $\theta$ is the parameter. Let $o_{u,i}$ be the indicator of whether user $u$ rated item $i$, and $\mathcal{O} = \{(u,i) \in \mathcal{D} | o_{u,i} = 1\}$ be the user-item index set with observed ratings, and $\mathbf{R}^o = \{r_{u,i} \in \mathbf{R} | o_{u,i} = 1\}$ be the observed ratings. If $\mathbf{R}$ is fully observed, then the prediction model $f(x_{u,i}; \theta)$ can be trained by minimizing the ideal loss function

$$\mathcal{L}_{ideal}(\theta) = \frac{1}{|\mathcal{D}|} \sum_{(u,i) \in \mathcal{D}} \delta_{u,i},$$

where $\delta_{u,i} = r_{u,i} \delta^{(1)}(\hat{r}_{u,i}) + (1 - r_{u,i}) \delta^{(0)}(\hat{r}_{u,i})$ is the prediction error, and $\delta^{(r)}(\hat{r}_{u,i})$ is a pre-defined loss function for $r = 0, 1$. For example, $\delta^{(r)}(\hat{r}_{u,i}) = -r \log \hat{r}_{u,i} - (1 - r) \log (1 - \hat{r}_{u,i})$ represents the cross-entropy loss. However, optimizing the ideal loss is infeasible, as $r_{u,i}$ is observable only when $o_{u,i} = 1$. A naive method is to optimize the prediction model directly using the user-item pairs with observed ratings, but this will incurs sample selection bias because the user-item pairs with observed ratings are no longer representative of all user-item pairs.

To address this problem, many debiasing methods have been proposed by designing unbiased estimators of the ideal loss. For example, the EIB method directly imputes the label $r_{u,i}$ corresponding to missing events, with the estimator

$$\mathcal{L}_{\text{EIB}}(\theta) = \frac{1}{|\mathcal{D}|} \sum_{(u,i) \in \mathcal{D}} \left[ o_{u,i} \delta_{u,i} + (1 - o_{u,i}) \hat{\delta}_{u,i} \right],$$

where $\hat{\delta}_{u,i} = \tilde{r}_{u,i} \delta^{(1)}(\hat{r}_{u,i}) + (1 - \tilde{r}_{u,i}) \delta^{(0)}(\hat{r}_{u,i})$ is the imputed error, $\tilde{r}_{u,i}$ is the pseudo-labeling for estimating $r_{u,i}$ given by a labeling-imputation model. Clearly, $\mathcal{L}_{EIB}(\theta)$ is an unbiased estimator of the ideal loss when all the pseudo-labelings are accurate, i.e., $\tilde{r}_{u,i} = r_{u,i}$ for $(u,i) \in \mathcal{D} \setminus \mathcal{O}$. Nevertheless, the EIB method usually has sub-optimal performance in practice due to the difficulty of obtaining accurate pseudo-labelings for the missing ratings (Guo et al., 2021). By additionally introducing the propensity $p_{u,i} = \mathbb{P}(o_{u,i} = 1 | x_{u,i})$, the DR estimator is proposed as

$$\mathcal{L}_{\text{DR}}(\theta) = \frac{1}{|\mathcal{D}|} \sum_{(u,i) \in \mathcal{D}} \left[ \hat{\delta}_{u,i} + \frac{o_{u,i}(\delta_{u,i} - \hat{\delta}_{u,i})}{\hat{p}_{u,i}} \right],$$

where $\hat{p}_{u,i}$ is the propensity model for estimating $p_{u,i}$. Despite theoretically being doubly robust, i.e., unbiasedness holds when either the learned propensities or the pseudo-labelings are accurate for all user-item pairs, however, it has been widely shown that the DR estimator would result in severe bias under inaccurate pseudo-labelings if the learned propensities are slightly inaccurate (Tan, 2007; Kang and Schafer, 2007; Molenberghs et al., 2015; Seaman and Vansteelandt, 2018).

## 3 PROPOSED METHOD

In this section, we first propose User-DR, Item-DR, and User-Item-DR estimators in Section 3.1, and theoretically show the unbiasedness of the proposed estimators for arbitrary user-specific and item-specific inductive biases, which greatly weakens the unbiasedness condition of the previous DR estimators on pseudo-labelings. In Section 3.2, we show that the variances of the proposed estimators are highly controllable and manageable, provided that the reconstructed propensities do not differ much from the original propensities. In Section 3.3, we further propose a propensity reconstruction learning approach to effectively achieve unbiased learning.

### 3.1 USER-DR, ITEM-DR, AND USER-ITEM-DR ESTIMATORS

In contrast to previous DR estimators that directly use $\hat{p}_{u,i}$ as propensities, where $\hat{p}_{u,i}$ are obtained by performing a binary classification on $o_{u,i}$ using $x_{u,i}$ (Wang et al., 2019; Saito, 2020; Guo et al., 2021),

given a prediction model $\hat{r}_{u,i}$, the proposed User-DR estimator first learns a constrained propensity model $\tilde{p}_{u,i}$ that satisfies

$$\sum_{i \in \mathcal{I}} \left( \frac{o_{u,i}}{\tilde{p}_{u,i}} - 1 \right) \left( \delta^{(1)}(\hat{r}_{u,i}) - \delta^{(0)}(\hat{r}_{u,i}) \right) = 0, \quad \text{for all} \quad u \in \mathcal{U}, \tag{1}$$

where $\delta^{(r)}(\hat{r}_{u,i})$ is a pre-defined loss function for $r = 0, 1$, such as the cross-entropy loss $\delta^{(r)}(\hat{r}_{u,i}) = -r \log \hat{r}_{u,i} - (1 - r) \log (1 - \hat{r}_{u,i})$. Then the User-DR estimator is

$$\mathcal{L}_{\text{UDR}}(\theta) = \frac{1}{|\mathcal{D}|} \sum_{(u,i) \in \mathcal{D}} \left[ \hat{\delta}_{u,i} + \frac{o_{u,i}(\delta_{u,i} - \hat{\delta}_{u,i})}{\tilde{p}_{u,i}} \right],$$

which adopts a similar form to the DR estimator, but requiring the learned propensities $\tilde{p}_{u,i}$ satisfies the above constraints in Eq. (1).

Now, we prove that the constraints in Eq. (1) can effectively alleviate the inaccurate pseudo-labelings problem in the previous DR estimators. Formally, the bias of User-DR estimator is

$$\text{Bias}(\mathcal{L}_{\text{UDR}}(\theta)) = \mathcal{L}_{ideal}(\theta) - \mathbb{E}(\mathcal{L}_{\text{UDR}}(\theta))$$

$$= \frac{1}{|\mathcal{D}|} \sum_{(u,i) \in \mathcal{D}} \delta_{u,i} - \frac{1}{|\mathcal{D}|} \sum_{(u,i) \in \mathcal{D}} \mathbb{E} \left[ \hat{\delta}_{u,i} + \frac{o_{u,i}(\delta_{u,i} - \hat{\delta}_{u,i})}{\tilde{p}_{u,i}} \right]$$

$$= \mathbb{E} \left[ \frac{1}{|\mathcal{D}|} \sum_{u \in \mathcal{U}} \sum_{i \in \mathcal{I}} \left( \frac{o_{u,i}}{\tilde{p}_{u,i}} - 1 \right) (\hat{\delta}_{u,i} - \delta_{u,i}) \right]$$

$$= \mathbb{E} \left[ \frac{1}{|\mathcal{D}|} \sum_{u \in \mathcal{U}} \sum_{i \in \mathcal{I}} \left( \frac{o_{u,i}}{\tilde{p}_{u,i}} - 1 \right) \left\{ \left( \delta^{(1)}(\hat{r}_{u,i}) - \delta^{(0)}(\hat{r}_{u,i}) \right) (\tilde{r}_{u,i} - r_{u,i}) \right\} \right].$$

The last equation holds directly from the definitions of $\hat{\delta}_{u,i}$ and $\delta_{u,i}$. On the one hand, similar to the previous DR estimators, the User-DR estimator is unbiased under either accurate pseudo-labelings $\tilde{r}_{u,i} = r_{u,i}$ or accurate learned propensities $\tilde{p}_{u,i} = p_{u,i} = \mathbb{P}(o_{u,i} = 1 | x_{u,i})$ for all user-item pairs. On the other hand, when the pseudo-labelings model has a user-specific inductive bias, i.e., $\tilde{r}_{u,i} = r_{u,i} + b_u$, multiplying both sides of the Eq. (1) by $b_u$ and summing over all $u$ yields

$$\text{Bias}(\mathcal{L}_{\text{UDR}}(\theta)) = \mathbb{E} \left[ \sum_{u \in \mathcal{U}} \sum_{i \in \mathcal{I}} \left( \frac{o_{u,i}}{\tilde{p}_{u,i}} - 1 \right) \cdot \left( \delta^{(1)}(\hat{r}_{u,i}) - \delta^{(0)}(\hat{r}_{u,i}) \right) \cdot b_u \right] = 0.$$

We summarize and compare the unbiasedness conditions of previous DR estimators (Wang et al., 2019; Saito, 2020; Zhang et al., 2020; Guo et al., 2021; Wang et al., 2022) and the proposed User-DR estimator in Lemma 1 and Theorem 1, respectively.

**Lemma 1** (Wang et al. (2019)). *The DR estimator is unbiased when either **pseudo-labelings are accurate** $\tilde{r}_{u,i} = r_{u,i}$ or learned propensities are accurate $\hat{p}_{u,i} = p_{u,i}$ for all user-item pairs.*

**Theorem 1.** *The User-DR estimator is unbiased when either **pseudo-labelings deviate from the true labels with arbitrary user-specific inductive bias** $\tilde{r}_{u,i} = r_{u,i} + b_u$ or learned propensities are accurate $\tilde{p}_{u,i} = p_{u,i}$ for all user-item pairs.*

Empirically, it is meaningful to weaken the unbiasedness condition of the previous DR estimators to inaccurate pseudo-labelings with user-specific inductive bias. One scenario is the *user conformity effect* (Zheng et al., 2021; Chen et al., 2022), e.g., in the movie rating, a user may give a higher rating for a movie simply because the public gives it a high rating, ignoring such influence will make $\tilde{r}_{u,i}$ overestimates $r_{u,i}$. Another scenario is *user self-selection* (Schnabel et al., 2016; Liu et al., 2022; Lin et al., 2023), since each user may have a specific preference to items.

Since the improved theoretical guarantees for User-DR originate from constraints in Eq. (1), one may argue whether such constraints are too strong to be satisfied. In fact, a key observation is that when the learned propensities are accurate, i.e., $\tilde{p}_{u,i} = p_{u,i}$, then these constraints will be satisfied naturally, and $\mathcal{L}_{\text{UDR}}(\theta)$ will degenerates to $\mathcal{L}_{\text{DR}}(\theta)$, which does not impose additional constraints to reduce the accuracy of learned propensities. In contrast, if the learned propensities are inaccurate, i.e.,

$\tilde{p}_{u,i} \neq p_{u,i}$, the bias of the previous DR estimators will strictly depend on the accuracy of the pseudo-labelings, whereas the proposed User-DR reduces the influence of those inaccurate pseudo-labelings by learning an alternative propensity model that satisfies constraints in Eq. (1).

Similar to the construction of the User-DR estimator, we propose the Item-DR estimator that is unbiased to item-specific inductive bias by replacing the constraints in Eq. (1) with

$$\sum_{u \in \mathcal{U}} \left(\frac{o_{u,i}}{\tilde{p}_{u,i}} - 1\right) \left(\delta^{(1)}(\hat{r}_{u,i}) - \delta^{(0)}(\hat{r}_{u,i})\right) = 0, \quad \text{for all} \quad i \in \mathcal{I}. \tag{2}$$

Furthermore, the User-Item-DR estimator, which is robust to both user-specific and item-specific inductive biases, can be obtained by learning a propensity model satisfying constraints in both Eq. (1) and Eq. (2). Following a similar argument in Theorem 1, we have the following results.

**Corollary 1.** *(a) The Item-DR estimator is unbiased, if either (i) $\tilde{r}_{u,i} = r_{u,i} + b_i$, or (ii) $\tilde{p}_{u,i} = p_{u,i}$;*

*(b) The User-Item-DR estimator is unbiased, if either (i) $\tilde{r}_{u,i} = r_{u,i} + b_u + b_i$, or (ii) $\tilde{p}_{u,i} = p_{u,i}$.*

It is also meaningful to consider inaccurate pseudo-labelings with item-specific inductive bias, e.g., item popularity bias and item exposure position bias (see Appendix B for more discussions). Notably, we would like to clarify that *even if the "user/item-specific inductive bias" conditions are not strictly satisfied, the biases of the proposed DR estimators are still strictly smaller than the previous DR,* as long as the existence of a user-specific constant $b_u$ or an item-specific constant $b_i$ such that the bias arises from the inaccurate pseudo-labelings $\{\tilde{r}_{u,i} : i \in \mathcal{I}\}$ or $\{\tilde{r}_{u,i} : u \in \mathcal{U}\}$ can be reduced. We illustrate this with a toy example as follows. Suppose the inductive biases of the $\tilde{r}_{u,i}$ for user $u$ on items $i_1$, $i_2$, and $i_3$ are $\tilde{r}_{u,i_1} - r_{u,i_1} = 1$, $\tilde{r}_{u,i_2} - r_{u,i_2} = 2$, and $\tilde{r}_{u,i_3} - r_{u,i_3} = 3$, respectively. Then the UDR estimator are able to cancel a user-specific constant $b_u$ (e.g., $b_u = 2$) to make the inductive biases become $1 - b_u$, $2 - b_u$, and $3 - b_u$, which leads to smaller biases. Table 1 summarizes the advantages of the proposed DR estimators over the previous studies (Steck, 2010; Schnabel et al., 2016; Wang et al., 2019), which clearly relaxes the unbiasedness conditions on accurate pseudo-labelings, and provides a new research direction for debiased recommendations.

### 3.2 FURTHER THEORETICAL ANALYSIS ON VARIANCE

The proposed estimators greatly enhances the robustness of DR to inaccurate pseudo-labelings. A further question is whether such unbiasedness come at the cost of increased variance. Impressively, the variance are highly controllable and manageable as shown below (see Appendix C for proof).

**Theorem 2.** *If $1/L \leq \hat{p}_{u,i}^2/\tilde{p}_{u,i}^2 \leq L$ for a constant $L$,*

$$\frac{1}{L} \cdot \mathbb{V}(\mathcal{L}_{\text{DR}}(\theta)) \leq \mathbb{V}(\mathcal{L}_{\text{UDR}}(\theta)) \leq L \cdot \mathbb{V}(\mathcal{L}_{\text{DR}}(\theta)).$$

Theorem 2 shows that the variance of the User-DR estimator[1] can be controlled by the distance between the base propensities $\hat{p}_{u,i}$ and the learned constrained propensities $\tilde{p}_{u,i}$, which is essentially a bias-variance trade-off compared with the previous DR estimators. This motivates us to further propose a propensity reconstrction learning approach to meet the constraints in Eq. (1) and Eq. (2) with minimal changes to the original propensities $\hat{p}_{u,i}$ in the following Section 3.3.

### 3.3 PROPENSITY RECONSTRUCTION LEARNING

We next propose a propensity reconstruction learning approach that adaptively updates the constraint weights to meet the constraints of the proposed User-Item-DR estimator[2]. The proposed algorithm alternately trains a reconstructed propensity model, a pseudo-labeling model for imputing the prediction errors $\hat{\mathbf{E}} = \{\hat{\delta}_{u,i}|(u,i) \in \mathcal{D}\}$, and a rating prediction model $\hat{\mathbf{R}} = \{\hat{r}_{u,i}|(u,i) \in \mathcal{D}\}$.

**Step 1. Propensity Reconstruction $\hat{\mathbf{P}} \to \tilde{\mathbf{P}}$.** Given the learned propensities $\hat{\mathbf{P}} = \{\hat{p}_{u,i}|(u,i) \in \mathcal{D}\}$ without constraints, Theorem 2 states the distance between $\hat{\mathbf{P}}$ and $\tilde{\mathbf{P}} = \{\tilde{p}_{u,i}|(u,i) \in \mathcal{D}\}$ can provide an upper bound on the variance of the proposed User-Item-DR estimator. Therefore, a natural

---

[1] Theorem 2 also holds for Item-DR and User-Item-DR estimators from similar arguments.

[2] Without loss of generality, we use User-Item-DR estimator in Section 3.3 for illustration purpose.

idea is to reconstruct $\hat{\mathbf{P}}$ to the nearest $\tilde{\mathbf{P}}$ that satisfies the constraints in Eq. (1) and Eq. (2) in the User-Item-DR estimator. Formally, the optimization problem is

$$\min_{\tilde{p}} \sum_{u \in \mathcal{U}} \sum_{i \in \mathcal{I}} \left( \frac{1}{\hat{p}_{u,i}} - \frac{1}{\tilde{p}_{u,i}} \right)^2,$$

$$\text{s.t. } \tilde{p}_{u,i} > 0, \quad (u,i) \in \mathcal{D},$$

$$\sum_{i \in \mathcal{I}} \left( \frac{o_{u,i}}{\tilde{p}_{u,i}} - 1 \right) \left( \delta^{(1)}(\hat{r}_{u,i}) - \delta^{(0)}(\hat{r}_{u,i}) \right) = 0, \ u \in \mathcal{U},$$

$$\sum_{u \in \mathcal{U}} \left( \frac{o_{u,i}}{\tilde{p}_{u,i}} - 1 \right) \left( \delta^{(1)}(\hat{r}_{u,i}) - \delta^{(0)}(\hat{r}_{u,i}) \right) = 0, \ i \in \mathcal{I},$$

which is a convex optimization problem with respect to $1/\tilde{p}$. The following states the rationality of reconstructing the inverse of the propensities rather than the propensities themselves: first, the former leads to a convex optimization, so that gradient-based algorithms can efficiently find globally optimal solutions; second, from the theoretical analysis of DR estimators (Wang et al., 2019; Guo et al., 2021; Dai et al., 2022), the bias and variance of the DR estimators are proportional to the inverse propensities and squared inverse propensities, respectively, therefore providing more theoretical guarantees. In practice, the optimization problem can be solved by minimizing the reconstruction loss with the constraints as the regularizations that

$$\mathcal{L}(\tilde{p} \mid \hat{p}) = \frac{1}{2} \sum_{u \in \mathcal{U}} \sum_{i \in \mathcal{I}} \left( \frac{1}{\hat{p}_{u,i}} - \frac{1}{\tilde{p}_{u,i}} \right)^2 + \frac{\gamma}{2} \sum_{u \in \mathcal{U}} \lambda_u \left[ \sum_{i \in \mathcal{I}} \left( \frac{o_{u,i}}{\tilde{p}_{u,i}} - 1 \right) \left( \delta^{(1)}(\hat{r}_{u,i}) - \delta^{(0)}(\hat{r}_{u,i}) \right) \right]^2$$

$$+ \frac{\gamma}{2} \sum_{i \in \mathcal{I}} \lambda_i \left[ \sum_{u \in \mathcal{U}} \left( \frac{o_{u,i}}{\tilde{p}_{u,i}} - 1 \right) \left( \delta^{(1)}(\hat{r}_{u,i}) - \delta^{(0)}(\hat{r}_{u,i}) \right) \right]^2,$$

where $\tilde{p}_{u,i} = \pi(x_{u,i}; \alpha)$ is the reconstructed propensity model, $\lambda_u$ and $\lambda_i$ are Lagrange multipliers reflecting the constraint strength, $\gamma$ is a trade-off hyper-parameter. Nevertheless, there are $O(|\mathcal{U}|+|\mathcal{I}|)$ constraints as well as the Lagrange multipliers in Eq. (1) and Eq. (2). Therefore, the dual optimization will not lead to faster efficiency. To address this problem, we propose an attention mechanism for collaborative filtering that adaptively learns the constraint strength (which is also considered to be the role of Lagrange multipliers), thus reducing the number of parameters of the dual problem. Specifically, let $\boldsymbol{s}_u$ and $\boldsymbol{t}_i$ be the latent vectors of user $u$ and item $i$, we propose to use an attention mechanism to learn $\lambda_u$ and $\lambda_i$, which can be formalized as

$$\lambda_u = \frac{\sum_{i \in \mathcal{I}} \exp(\tilde{\boldsymbol{s}}_u^\top \boldsymbol{t}_i)}{\sum_{u \in \mathcal{U}} \sum_{i \in \mathcal{I}} \exp(\tilde{\boldsymbol{s}}_u^\top \boldsymbol{t}_i)}, \quad \text{and} \quad \tilde{\boldsymbol{s}}_u = \tanh(\boldsymbol{A} \boldsymbol{s}_u + \boldsymbol{b}),$$

where $\boldsymbol{A}$ is the connection weight matrix and $\boldsymbol{b}$ is the bias, and $\lambda_i$ can be obtained from a similar way. We further empirically explored other selections of $\lambda_u$ in Section 5, such as the constant weights $\lambda_u = \lambda = 1/|\mathcal{U}|$, or obtain the weights via a multilayer perceptron.

**Step 2. Training Pseudo-labelling with $\tilde{\mathbf{P}}$.** The pseudo-labeling model can be learned by minimizing the weighted average loss of the prediction error and imputed error of the observed samples

$$\mathcal{L}_e(\beta; \alpha, \theta \mid \tilde{\mathbf{P}}) = \frac{1}{|\mathcal{D}|} \sum_{(u,i) \in \mathcal{D}} \frac{o_{u,i}(\delta_{u,i} - \hat{\delta}_{u,i})^2}{\tilde{p}_{u,i}},$$

where $\beta$ is the parameter of the pseudo-labeling model, $\delta_{u,i} = r_{u,i}\delta^{(1)}(\hat{r}_{u,i}) + (1 - r_{u,i})\delta^{(0)}(\hat{r}_{u,i})$ is the prediction error, and $\hat{\delta}_{u,i} = \tilde{r}_{u,i}\delta^{(1)}(\hat{r}_{u,i}) + (1 - \tilde{r}_{u,i})\delta^{(0)}(\hat{r}_{u,i})$ is the imputed error.

**Step 3. Training Prediction Model with $\tilde{\mathbf{P}}$.** Given the reconstructed propensities $\tilde{p}_{u,i}$ obtained in Step 1, the prediction model can be learned by minimizing the proposed User-Item-DR loss

$$\mathcal{L}_r(\theta; \alpha, \beta \mid \tilde{\mathbf{P}}) = \frac{1}{|\mathcal{D}|} \sum_{(u,i) \in \mathcal{D}} \left[ \hat{\delta}_{u,i} + \frac{o_{u,i}(\delta_{u,i} - \hat{\delta}_{u,i})}{\tilde{p}_{u,i}} \right].$$

By alternately implementing the above steps, the prediction model can achieve debiased learning under inaccurate pseudo-labelings. We summarize the alternating training process in Appendix D.

Table 2: Relative errors on ML-100K dataset with user-specific and item-specific inductive bias.

| U-Bias | Naive | EIB | IPS | SNIPS | DR | UDR | IDR | UIDR |
|---|---|---|---|---|---|---|---|---|
| ONE | 0.068 ± 0.003 | 0.223 ± 0.004 | 0.035 ± 0.004 | 0.034 ± 0.004 | 0.047 ± 0.005 | **0.006 ± 0.006*** | **0.016 ± 0.011*** | **0.003 ± 0.002*** |
| THREE | 0.078 ± 0.004 | 0.234 ± 0.004 | 0.040 ± 0.004 | 0.039 ± 0.005 | 0.049 ± 0.005 | **0.005 ± 0.003*** | **0.017 ± 0.028*** | **0.002 ± 0.001*** |
| FIVE | 0.100 ± 0.004 | 0.247 ± 0.004 | 0.050 ± 0.004 | 0.050 ± 0.005 | 0.054 ± 0.005 | **0.009 ± 0.009*** | **0.028 ± 0.025*** | **0.010 ± 0.006*** |
| ROTATE | 0.137 ± 0.002 | 0.036 ± 0.001 | 0.068 ± 0.004 | 0.069 ± 0.002 | 0.008 ± 0.002 | **0.001 ± 0.001*** | **0.003 ± 0.003*** | **0.002 ± 0.001*** |
| SKEW | 0.025 ± 0.002 | 0.108 ± 0.002 | 0.012 ± 0.002 | **0.012 ± 0.002** | 0.028 ± 0.003 | **0.003 ± 0.002*** | 0.014 ± 0.015 | **0.002 ± 0.001*** |
| CRS | 0.105 ± 0.003 | 0.216 ± 0.004 | 0.053 ± 0.003 | 0.052 ± 0.004 | 0.024 ± 0.003 | **0.004 ± 0.005*** | **0.012 ± 0.013*** | **0.003 ± 0.000*** |

| I-Bias | Naive | EIB | IPS | SNIPS | DR | UDR | IDR | UIDR |
|---|---|---|---|---|---|---|---|---|
| ONE | 0.069 ± 0.004 | 0.222 ± 0.003 | 0.034 ± 0.004 | 0.035 ± 0.005 | 0.049 ± 0.005 | **0.031 ± 0.012** | **0.010 ± 0.007*** | **0.004 ± 0.002*** |
| THREE | 0.078 ± 0.003 | 0.234 ± 0.004 | 0.038 ± 0.003 | 0.039 ± 0.004 | 0.050 ± 0.004 | **0.017 ± 0.013*** | **0.012 ± 0.020*** | **0.006 ± 0.003*** |
| FIVE | 0.103 ± 0.004 | 0.245 ± 0.005 | 0.050 ± 0.004 | 0.052 ± 0.004 | 0.057 ± 0.004 | **0.013 ± 0.012*** | **0.012 ± 0.008*** | **0.007 ± 0.003*** |
| ROTATE | 0.138 ± 0.004 | 0.035 ± 0.001 | 0.070 ± 0.004 | 0.069 ± 0.003 | 0.008 ± 0.001 | **0.002 ± 0.002*** | **0.001 ± 0.000*** | **0.002 ± 0.001*** |
| SKEW | 0.025 ± 0.003 | 0.106 ± 0.001 | 0.011 ± 0.002 | 0.012 ± 0.003 | 0.028 ± 0.002 | **0.009 ± 0.006*** | **0.009 ± 0.003*** | **0.004 ± 0.001*** |
| CRS | 0.105 ± 0.004 | 0.216 ± 0.002 | 0.051 ± 0.003 | 0.052 ± 0.004 | **0.024 ± 0.004** | 0.031 ± 0.019 | **0.008 ± 0.006*** | **0.006 ± 0.003*** |

| UI-Bias | Naive | EIB | IPS | SNIPS | DR | UDR | IDR | UIDR |
|---|---|---|---|---|---|---|---|---|
| ONE | 0.066 ± 0.001 | 0.445 ± 0.006 | **0.031 ± 0.002** | 0.032 ± 0.002 | 0.094 ± 0.003 | 0.062 ± 0.011 | **0.025 ± 0.023** | **0.007 ± 0.007*** |
| THREE | 0.076 ± 0.002 | 0.470 ± 0.004 | **0.036 ± 0.003** | 0.037 ± 0.003 | 0.099 ± 0.003 | 0.050 ± 0.032 | 0.062 ± 0.063 | **0.009 ± 0.003*** |
| FIVE | 0.099 ± 0.001 | 0.488 ± 0.003 | 0.047 ± 0.002 | 0.048 ± 0.002 | 0.109 ± 0.001 | **0.037 ± 0.028*** | **0.013 ± 0.013*** | **0.009 ± 0.006** |
| ROTATE | 0.138 ± 0.001 | 0.071 ± 0.002 | 0.070 ± 0.002 | 0.069 ± 0.002 | 0.015 ± 0.001 | **0.004 ± 0.004*** | **0.007 ± 0.010*** | **0.002 ± 0.001*** |
| SKEW | 0.025 ± 0.001 | 0.214 ± 0.003 | **0.011 ± 0.001** | 0.012 ± 0.001 | 0.056 ± 0.002 | 0.015 ± 0.011 | 0.027 ± 0.030 | **0.012 ± 0.004** |
| CRS | 0.105 ± 0.003 | 0.432 ± 0.003 | 0.051 ± 0.003 | 0.052 ± 0.003 | 0.048 ± 0.001 | **0.048 ± 0.035** | **0.013 ± 0.007*** | **0.009 ± 0.008*** |

Note: * means statistically significant results (p-value $\leq$ 0.05) using the paired-t-test compared with the best baseline. We bold the best three results and underline the best baseline result.

## 4 SEMI SYNTHETIC EXPERIMENTS

**Experiment Setup.** We conduct semi synthetic experiments using MOVIELENS 100K (ML-100K) dataset, which contains 100,000 ratings from 943 users for 1,682 items. We focus on two research questions below: (1) whether the proposed estimators are unbiased in the presence of user-specific and item-specific inductive bias; (2) how the varying bias level affects the estimators' performance.

**Experimental Details.** We first generate the ground truth probability matrix $\mathbf{R}$, ground truth propensity matrix $\mathbf{P}$ and observation matrix $\mathbf{O}$ following the previous studies (Schnabel et al., 2016; Wang et al., 2019; Guo et al., 2021) (see Appendix E for the generation process details). To verify the effectiveness of the proposed estimators, we generate several $\hat{\mathbf{R}}$ based on $\mathbf{R}$ as follows:

• **ONE**: The predicted matrix $\hat{\mathbf{R}}$ is identical to the true matrix $\mathbf{R}$, except that randomly select $r_{u,i} = 0.1$ with total amount $|\{(u, i) \mid r_{u,i} = 0.9\}|$ are flipped to 0.9.
• **THREE**: Same as **ONE**, but flipping $r_{u,i} = 0.3$ instead.
• **FIVE**: Same as **ONE**, but flipping $r_{u,i} = 0.5$ instead.
• **ROTATE**: $\hat{r}_{u,i} = r_{u,i} - 0.2$ when $r_{u,i} \geq 0.3$, and $\hat{r}_{u,i} = 0.9$ when $r_{u,i} = 0.1$.
• **SKEW**: Predicted $\hat{r}_{u,i}$ are sampled from the Gaussian distribution $\mathcal{N}(\mu = r_{u,i}, \sigma = (1 - r_{u,i})/2)$, and clipped to the interval $[0.1, 0.9]$.
• **CRS**: If the true $r_{u,i} \leq 0.6$, then $\hat{r}_{u,i} = 0.2$. Otherwise, $\hat{r}_{u,i} = 0.6$.

Following the previous studies (Guo et al., 2021; Dai et al., 2022), we estimate the propensity by $1/\hat{p}_{u,i} = (1 - \rho)/p_{u,i} + \rho/p_e$, where $p_{u,i}$ is the ground truth propensity, $p_e = |\mathcal{D}|^{-1} \sum_{(u,i)\in\mathcal{D}} o_{u,i}$, and $\rho$ is randomly sampled from the uniform distribution $U(0, 1)$ to introduce noises. Next, we simulate the biased pseudo-labelings $\tilde{r}_{u,i}$ for EIB, DR and the proposed estimators in three ways: (1) $\tilde{r}_{u,i} = r_{u,i} + b_u$ (user-specific inductive bias); (2) $\tilde{r}_{u,i} = r_{u,i} + b_i$ (item-specific inductive bias); (3) $\tilde{r}_{u,i} = r_{u,i} + b_u + b_i$ (user-item inductive bias), where $b_u$ and $b_i$ are randomly sampled from the uniform distribution $U(0, \nu)$ to generate inaccurate pseudo-labelings. Finally we use $r_{u,i} \in [0, 1]$ as the positive sample probabilities for Bernoulli sampling to obtain the binary rating matrix. The absolute relative error (RE) is used for evaluation, which is defined as $\text{RE}(\mathcal{L}_{est}) = |\mathcal{L}_{ideal}(\hat{\mathbf{R}}) - \mathcal{L}_{est}(\hat{\mathbf{R}})|/\mathcal{L}_{ideal}(\hat{\mathbf{R}})$, where $\mathcal{L}_{est}$ denotes the loss of estimator to be compared. RE evaluates the accuracy of the estimated loss, the smaller the RE, the more accurate the estimation.

**Performance Comparison**. The experiment results with bias level $\nu = 0.2$ are shown in Table 2. First, the proposed estimators significantly outperform the baselines in all scenarios. Impressively, the proposed User-DR (or Item-DR) estimator is still able to outperform previous DR estimator in item-specific (or user-specific) inductive bias scenarios. Second, the User-DR (or the Item-DR) outperforms in the case of user-specific (or item-specific) inductive bias, and the User-Item-DR estimator shows the most competing performance in all three scenarios, which further validates the ability to eliminate the corresponding types of inductive bias. Meanwhile, since the IPS and SNIPS estimators are not affected by pseudo-labelings, they show competitive performance in a few scenarios when increasing the inductive biases of the pseudo-labelings.

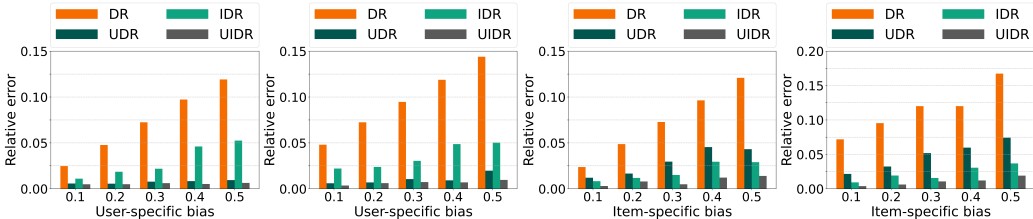

Figure 1: Relative error with varying inductive bias size. The left (right) two figures are user(item)-specific inductive bias scenario with the item(user)-specific inductive bias fixed as 0 and 0.1.

Table 3: Performance on AUC, NDCG@K, and F1@K on the unbiased test set of Music and KuaiRec. The best three results are bolded, and the best baseline is underlined.

| Method | COAT | | | MUSIC | | | KUAIREC | | |
|---|---|---|---|---|---|---|---|---|---|
| | AUC | N@5 | F1@5 | AUC | N@5 | F1@5 | AUC | N@50 | F1@50 |
| MF | $0.680_{\pm0.006}$ | $0.616_{\pm0.011}$ | $0.470_{\pm0.006}$ | $0.651_{\pm0.005}$ | $0.626_{\pm0.001}$ | $0.300_{\pm0.001}$ | $0.741_{\pm0.003}$ | $0.724_{\pm0.003}$ | $0.566_{\pm0.002}$ |
| IPS | $0.710_{\pm0.003}$ | $0.603_{\pm0.009}$ | $0.450_{\pm0.008}$ | $0.656_{\pm0.002}$ | $0.633_{\pm0.001}$ | $0.308_{\pm0.001}$ | $0.750_{\pm0.003}$ | $0.734_{\pm0.003}$ | $0.572_{\pm0.002}$ |
| ASIPS | $0.712_{\pm0.008}$ | $0.627_{\pm0.010}$ | $0.470_{\pm0.007}$ | $0.661_{\pm0.003}$ | $0.641_{\pm0.004}$ | $0.322_{\pm0.003}$ | $0.746_{\pm0.009}$ | $0.733_{\pm0.004}$ | $0.585_{\pm0.006}$ |
| DR | $0.710_{\pm0.006}$ | $0.632_{\pm0.003}$ | $0.471_{\pm0.003}$ | $0.656_{\pm0.009}$ | $0.669_{\pm0.007}$ | $0.330_{\pm0.005}$ | $0.745_{\pm0.004}$ | $0.718_{\pm0.003}$ | $0.574_{\pm0.003}$ |
| DR-JL | $0.714_{\pm0.007}$ | $0.646_{\pm0.009}$ | $0.486_{\pm0.006}$ | $0.682_{\pm0.001}$ | $0.660_{\pm0.002}$ | $0.326_{\pm0.001}$ | $0.759_{\pm0.002}$ | $0.757_{\pm0.004}$ | $0.582_{\pm0.005}$ |
| MRDR-JL | $0.715_{\pm0.004}$ | $0.653_{\pm0.006}$ | $0.492_{\pm0.005}$ | $0.684_{\pm0.001}$ | $0.645_{\pm0.001}$ | $0.315_{\pm0.001}$ | $0.762_{\pm0.003}$ | $0.751_{\pm0.002}$ | $0.579_{\pm0.003}$ |
| CVIB | $0.718_{\pm0.004}$ | $0.640_{\pm0.008}$ | $0.486_{\pm0.008}$ | $0.685_{\pm0.001}$ | $0.647_{\pm0.003}$ | $0.316_{\pm0.002}$ | $0.758_{\pm0.001}$ | $0.752_{\pm0.001}$ | $0.575_{\pm0.001}$ |
| DIB | $0.726_{\pm0.003}$ | $0.628_{\pm0.008}$ | $0.469_{\pm0.007}$ | $0.690_{\pm0.002}$ | $0.653_{\pm0.002}$ | $0.320_{\pm0.001}$ | $0.775_{\pm0.001}$ | $0.760_{\pm0.001}$ | $0.592_{\pm0.001}$ |
| DR-BIAS | $0.725_{\pm0.004}$ | $0.643_{\pm0.008}$ | $0.474_{\pm0.008}$ | $0.687_{\pm0.004}$ | $0.659_{\pm0.004}$ | $0.331_{\pm0.005}$ | $0.775_{\pm0.001}$ | $0.761_{\pm0.002}$ | $0.593_{\pm0.001}$ |
| DR-MSE | $0.715_{\pm0.001}$ | $0.630_{\pm0.009}$ | $0.475_{\pm0.008}$ | $0.685_{\pm0.001}$ | $0.648_{\pm0.002}$ | $0.316_{\pm0.002}$ | $0.779_{\pm0.002}$ | $0.773_{\pm0.002}$ | $0.589_{\pm0.002}$ |
| MR | $0.728_{\pm0.006}$ | $0.655_{\pm0.009}$ | $0.489_{\pm0.007}$ | $0.698_{\pm0.004}$ | $\underline{0.680}_{\pm0.002}$ | $0.323_{\pm0.005}$ | $0.776_{\pm0.002}$ | $0.793_{\pm0.001}$ | $0.599_{\pm0.002}$ |
| TDR | $0.730_{\pm0.008}$ | $0.651_{\pm0.013}$ | $0.487_{\pm0.010}$ | $0.694_{\pm0.002}$ | $0.667_{\pm0.003}$ | $\underline{0.337}_{\pm0.002}$ | $0.792_{\pm0.004}$ | $0.799_{\pm0.003}$ | $0.604_{\pm0.003}$ |
| TDR-JL | $0.729_{\pm0.005}$ | $\underline{0.656}_{\pm0.011}$ | $\underline{0.493}_{\pm0.010}$ | $\mathbf{0.702}_{\pm0.002}$ | $0.672_{\pm0.004}$ | $0.332_{\pm0.003}$ | $\underline{0.793}_{\pm0.004}$ | $\underline{0.799}_{\pm0.004}$ | $0.603_{\pm0.003}$ |
| Stable-DR | $0.719_{\pm0.006}$ | $0.631_{\pm0.008}$ | $0.475_{\pm0.006}$ | $0.687_{\pm0.001}$ | $0.650_{\pm0.003}$ | $0.316_{\pm0.002}$ | $0.764_{\pm0.003}$ | $0.791_{\pm0.003}$ | $0.595_{\pm0.002}$ |
| ESMM | $0.686_{\pm0.004}$ | $0.638_{\pm0.005}$ | $0.485_{\pm0.004}$ | $0.601_{\pm0.002}$ | $0.665_{\pm0.003}$ | $0.328_{\pm0.001}$ | $0.721_{\pm0.006}$ | $0.764_{\pm0.006}$ | $0.576_{\pm0.004}$ |
| Multi-IPS | $0.711_{\pm0.005}$ | $0.604_{\pm0.005}$ | $0.463_{\pm0.008}$ | $0.651_{\pm0.005}$ | $0.667_{\pm0.006}$ | $0.331_{\pm0.004}$ | $0.748_{\pm0.005}$ | $0.738_{\pm0.006}$ | $0.579_{\pm0.004}$ |
| Multi-DR | $0.719_{\pm0.004}$ | $0.634_{\pm0.009}$ | $0.480_{\pm0.011}$ | $0.686_{\pm0.003}$ | $0.660_{\pm0.006}$ | $0.323_{\pm0.002}$ | $0.752_{\pm0.014}$ | $0.767_{\pm0.012}$ | $0.581_{\pm0.008}$ |
| ESCM$^2$-IPS | $0.721_{\pm0.005}$ | $0.645_{\pm0.006}$ | $0.490_{\pm0.005}$ | $0.680_{\pm0.002}$ | $0.653_{\pm0.002}$ | $0.322_{\pm0.002}$ | $0.779_{\pm0.001}$ | $0.767_{\pm0.001}$ | $0.592_{\pm0.002}$ |
| ESCM$^2$-DR | $\underline{0.730}_{\pm0.009}$ | $0.642_{\pm0.010}$ | $0.489_{\pm0.009}$ | $0.688_{\pm0.001}$ | $0.669_{\pm0.002}$ | $0.326_{\pm0.001}$ | $0.788_{\pm0.001}$ | $0.796_{\pm0.001}$ | $\underline{0.606}_{\pm0.001}$ |
| UDR (ours) | $\mathbf{0.739}^*_{\pm0.004}$ | $\mathbf{0.676}^*_{\pm0.003}$ | $\mathbf{0.521}^*_{\pm0.003}$ | $\mathbf{0.705}^*_{\pm0.001}$ | $\mathbf{0.766}^*_{\pm0.002}$ | $\mathbf{0.389}^*_{\pm0.001}$ | $\mathbf{0.802}^*_{\pm0.003}$ | $\mathbf{0.804}^*_{\pm0.002}$ | $\mathbf{0.610}^*_{\pm0.002}$ |
| IDR (ours) | $0.721_{\pm0.002}$ | $\mathbf{0.681}^*_{\pm0.003}$ | $\mathbf{0.529}^*_{\pm0.005}$ | $0.694_{\pm0.004}$ | $\mathbf{0.747}^*_{\pm0.002}$ | $\mathbf{0.378}^*_{\pm0.002}$ | $\mathbf{0.801}^*_{\pm0.002}$ | $\mathbf{0.803}^*_{\pm0.002}$ | $\mathbf{0.607}^*_{\pm0.002}$ |
| UIDR (ours) | $\mathbf{0.740}^*_{\pm0.006}$ | $\mathbf{0.722}^*_{\pm0.006}$ | $\mathbf{0.539}^*_{\pm0.005}$ | $\mathbf{0.713}^*_{\pm0.001}$ | $\mathbf{0.752}^*_{\pm0.001}$ | $\mathbf{0.382}^*_{\pm0.001}$ | $\mathbf{0.804}^*_{\pm0.004}$ | $\mathbf{0.804}^*_{\pm0.004}$ | $\mathbf{0.610}^*_{\pm0.003}$ |

Note: * means statistically significant results (p-value $\leq 0.05$) using the paired-t-test compared with the best baseline.

Moreover, we also conduct experiments when the unbiasedness conditions are violated, i.e., $\tilde{r}_{u,i} = r_{u,i} + b_{u,i}$, where $b_{u,i}$ is the inductive bias for more general cases, and the results are shown in the Appendix E to further validate the superiority of our methods. Figure 1 shows the experiment results with varying inductive bias levels by taking **ONE** for illustration, and similar results can be found in the other five prediction matrices. In the user-specific inductive bias scenario, as the bias level increases, both User-DR and User-Item-DR estimators demonstrate relatively stable performance. The Item-DR estimator exhibits a slow increase in REs, while the DR estimator experiences a rapid increase in such REs. Similar trends are observed in the item-specific inductive bias scenario.

## 5 REAL-WORLD EXPERIMENTS

**Dataset and Experimental Details.** Three widely-used real-world datasets are adopted in our experiments, which are COAT, MUSIC and KUAIREC (Gao et al., 2022). COAT contains ratings from 290 users to 300 items, with 6,960 biased ratings and 4,640 unbiased ratings in total. MUSIC contains ratings from 15,400 users to 1,000 items, with 311,704 biased ratings and 54,000 unbiased ratings. KUAIREC contains a fully exposed industrial dataset which contains 4,676,570 video watching ratio records from 1,411 users to 3,327 videos. Three widely-used evaluation metric, namely AUC, NDCG@K, and F1@K, are used to evaluate performance, where K is set to 5 for COAT and MUSIC and K is set to 50 for KUAIREC (see Appendix F for more experiment details).

**Baselines.** We take **Matrix Factorization (MF)** (Koren et al., 2009) as the base model. We compare the proposed methods with the following baselines: **IPS** (Saito et al., 2020; Schnabel et al., 2016), **ASIPS** (Saito, 2020), **DR** (Saito, 2020), **CVIB** (Wang et al., 2020b), **DIB** (Liu et al., 2021), **TDR** (Li et al., 2023b), **DR-BIAS** (Dai et al., 2022), **DR-MSE** (Dai et al., 2022), **Stable-DR** (Li et al., 2023c), and **MR** (Li et al., 2023a). We also consider the following baselines based on joint learning and multi-task learning: **DR-JL** (Wang et al., 2019), **MRDR-JL** (Guo et al., 2021), **TDR-JL** (Li et al., 2023b), **ESMM** (Ma et al., 2018), **Multi-IPS** (Zhang et al., 2020), **Multi-DR** (Zhang et al., 2020), **ESCM$^2$-IPS** (Wang et al., 2022) and **ESCM$^2$-DR** (Wang et al., 2022).

**Performance Analysis**. The results of proposed methods and baselines on COAT, MUSIC and KUAIREC are shown in Table 3. First, almost all debiasing methods perform better than the base model (MF), which shows the necessity of debiasing in RS. Second, all three proposed methods significantly outperform the competing baselines with p-value less than 0.05, which is attributed to the more relaxed and realistic unbiasedness conditions compared to the baseline DR methods. Finally, among three proposed methods, Item-DR performs slightly worse than User-DR and User-Item-DR, which reveals the inductive bias on the item-side is less than that on the user-side in practice.

**Ablation Studies**. We conduct the ablation study for User-DR on the large-scale industrial dataset KUAIREC. The results are shown in Table 4. When only the reconstruction loss is retained, our propensity model will be the same as the base propensity model. Therefore, User-DR degenerates to the DR-JL. Conversely, when only the constraint losses are retained, the reconstructed propensities will be considerably more distinct from the base propensities, leading to an increasing variance as shown in Theorem 2, which harms the prediction performance.

Table 4: Ablation study on KUAIREC.

| Method | $\mathcal{L}_{re}$ | $\gamma$ | AUC | N@50 | F1@50 |
|---|---|---|---|---|---|
| DR-JL | × | × | 0.759 | 0.757 | 0.582 |
| UDR w/o $\gamma$ | ✓ | × | 0.754 | 0.754 | 0.589 |
| UDR w/o $\mathcal{L}_{re}$ | × | ✓ | 0.787 | 0.789 | 0.599 |
| UDR | ✓ | ✓ | **0.802** | **0.804** | **0.610** |

Table 5: In-depth Analysis for $\lambda_u$.

| | MUSIC | | | KUAIREC | | |
|---|---|---|---|---|---|---|
| Method | AUC | N@5 | F1@5 | AUC | N@50 | F1@50 |
| Constant | 0.695 | 0.739 | 0.373 | 0.797 | 0.796 | 0.605 |
| MLP | 0.696 | 0.743 | 0.376 | 0.803 | 0.801 | 0.605 |
| Attention | **0.713** | **0.752** | **0.382** | **0.804** | **0.804** | **0.610** |

**Sensitivity Analysis**. The hyper-parameter $\gamma$ balances between the constraints to combat the inaccurate pseudo-labelings and the propensity reconstruction loss to control the variance. Specifically, a large value of $\gamma$ causes the reconstructed propensities to satisfy the constraints in Section 3, but may reduce the accuracy of the propensity estimation. We explore the effect of hyper-parameter $\gamma$ on the performance of User-Item-DR on two datasets MUSIC and KUAIREC, and the corresponding results are shown in Figure 2. We can see that moderate constraints are most helpful to trade-off between propensity estimation and estimator robustness and thus improve the debiased performance.

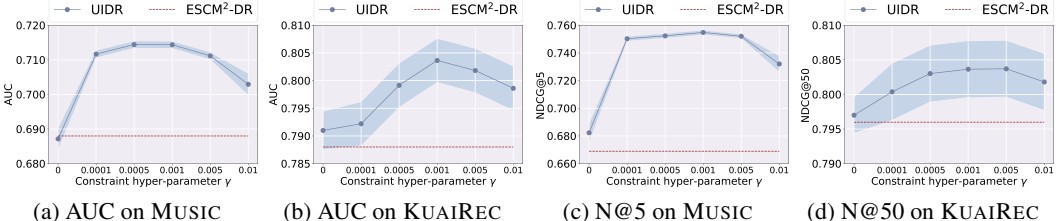

(a) AUC on MUSIC  (b) AUC on KUAIREC  (c) N@5 on MUSIC  (d) N@50 on KUAIREC

Figure 2: Effects of varying hyper-parameter $\gamma$ in UIDR loss.

**In-Depth Analysis**. The weight of each constraint is crucial throughout the learning process. Table 5 shows the impact of different models when learning $\lambda_u$ on User-DR prediction performance. When we use the constant model to generalize all the $\lambda_u$, each user's constraint is equally important. As a result, the model will not utilize the user information effectively, which harms the performance. In addition, although MLP has a good fitting capacity, it is easy to allocate a higher weight to some specific constraints and to ignore other constraints, which also harms the performance. Therefore, when the attention mechanism is used for model training, it allows the model to adaptively obtain fitting capacity and to prevent overfitting, which results in the most desirable performance.

## 6 CONCLUSION

In this paper, we proposed the User-DR, Item-DR, and User-Item-DR estimators that can achieve unbiased learning even under inaccurate pseudo-labelings. The proposed estimators greatly relax the unbiasedness condition and improve the robustness of existing DR estimators to inaccurate pseudo-labelings. Our theoretical analysis shows that the variance of the proposed estimators are highly controllable and manageable. We further propose a principled propensity reconstruction learning to be compatible with the theory of the proposed estimators, which uses an attention mechanism that adaptively updates the weights of the proposed constraints. Extensive experiments are conducted to verify the validity of our proposal. A limitation of this work is the proposed learning method cannot guarantee that all the constraints in Eq. (1) and Eq. (2) hold strictly, due to the computational burden, although this may lead to a more desirable debiasing performance.

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

## A    MORE DISCUSSIONS ON RELATED WORKS

Selection bias is a common issue recommendation datasets due to the self-selection behavior of users and the item selection process of the system (Chen et al., 2022; Wu et al., 2022). Error imputation-based (EIB) methods first learn pseudo-labelings for missing events and train the prediction model with both pseudo-labelings and observed labels (Marlin et al., 2007; Steck, 2013; Hernández-Lobato et al., 2014). However, the unbiasedness EIB requires that pseudo-labelings are accurate for all user-item pairs. By introducing an additional propensity model, doubly robust (DR) method is proposed to improve EIB (Wang et al., 2019), with many enhanced DR approaches developed (Guo et al., 2021; Zhang et al., 2020; Chen et al., 2021; Wang et al., 2021; 2022; Oosterhuis, 2023). The advantage of DR estimators is attributed to the property of double robustness, i.e., it is unbiased if either the pseudo-labelings or the learned propensities are accurate.

Despite the double robustness providing additional protection against inaccurate pseudo-labelings, i.e., unbiasedness holds when either the learned propensities or the pseudo-labelings are accurate for all user-item pairs, recent studies have shown that DR methods are highly sensitive to inaccurate pseudo-labelings, i.e., when the learned propensities are slightly inaccurate, the DR estimator can be severely biased with inaccurate pseudo-labelings (Kang and Schafer, 2007; Molenberghs et al., 2015; Vermeulen and Vansteelandt, 2015; Seaman and Vansteelandt, 2018). Thus, it can be summarized that the effectiveness of both the EIB and DR methods rely heavily on accurate pseudo-labelings.

Unfortunately, obtaining such accurate pseudo-labelings for all user-item pairs is usually impractical in practice, as user-item interactions are influenced by various factors, such as user self-selection (Ma et al., 2018; Luo et al., 2021), user conformity (Liu et al., 2016; Zheng et al., 2021), item popularity (Zhang et al., 2021; Wei et al., 2021), and item exposure position (Ai et al., 2018; Agarwal et al., 2019), causing inaccurate pseudo-labelings in practice. Motivated by this, we theoretically proposes several novel DR estimators and learning approach that achieve unbiasedness even with inaccurate pseudo-labelings, which greatly relaxes the unbiasedness condition of the doubly robust estimators.

## B    MORE DISCUSSIONS ON ITEM-SPECIFIC INDUCTIVE BIAS

In the following, we provide more discussions on the rationality of the item-specific inductive bias. For example, a recommender system algorithm that over-recommends popular items leads to *item popularity bias* (Zhang et al., 2021). In this case, the item-specific inductive bias arises from the influence of item popularity on the pseudo-labeling model. Another form of item-specific inductive bias may be caused by the *item exposure position* (Ai et al., 2018), and in such a case, the item-specific inductive bias is caused from the effect of item position on the learned pseudo-labelings.

## C    PROOF OF THEOREM 2

**Theorem 2.** *If* $1/L \leq \hat{p}_{u,i}^2 / \tilde{p}_{u,i}^2 \leq L$ *for a constant* $L$,

$$\frac{1}{L} \cdot \mathbb{V}(\mathcal{L}_{\mathrm{DR}}(\theta)) \leq \mathbb{V}(\mathcal{L}_{\mathrm{UDR}}(\theta)) \leq L \cdot \mathbb{V}(\mathcal{L}_{\mathrm{DR}}(\theta)).$$

*Proof.* As shown in Guo et al. (2021); Dai et al. (2022),

$$\mathbb{V}(\mathcal{L}_{\mathrm{DR}}(\theta)) = \frac{1}{|\mathcal{D}|^2} \sum_{(u,i) \in \mathcal{D}} \frac{p_{u,i}(1 - p_{u,i})(\delta_{u,i} - \hat{\delta}_{u,i})^2}{\hat{p}_{u,i}^2}.$$

The variance of UDR has the same form of DR but replaces $\hat{p}_{u,i}$ with $\tilde{p}_{u,i}$ satisfying Eq. (1),

$$\mathbb{V}(\mathcal{L}_{\mathrm{UDR}}(\theta)) = \frac{1}{|\mathcal{D}|^2} \sum_{(u,i) \in \mathcal{D}} \frac{p_{u,i}(1 - p_{u,i})(\delta_{u,i} - \hat{\delta}_{u,i})^2}{\tilde{p}_{u,i}^2}.$$

Now, it is clear that if $1/L \leq \hat{p}_{u,i}^2 / \tilde{p}_{u,i}^2 \leq L$ for a constant $L$, then we have

$$\frac{1}{L} \leq \frac{\mathbb{V}(\mathcal{L}_{\mathrm{UDR}}(\theta))}{\mathbb{V}(\mathcal{L}_{\mathrm{DR}}(\theta))} = \frac{\hat{p}_{u,i}^2}{\tilde{p}_{u,i}^2} \leq L.$$

$\square$

## D  PSEUDO-CODE FOR PROPENSITY RECONSTRUCTION LEARNING

We provide pseudo-code for the proposed propensity reconstruction learning in Section 3.3 below, where three steps (step 1: propensity reconstruction, step 2: training pseudo-labelling with $\tilde{\mathbf{P}}$, and step 3: training prediction model with $\tilde{\mathbf{P}}$) are alternatively implemented.

---

**Algorithm 1:** Propensity Reconstruction Learning

---

  **Input:** observed ratings $\mathbf{R}^o$ and learned propensities $\hat{\mathbf{P}}$
  **while** stopping criteria is not satisfied **do**
      **for** number of steps training the propensity model **do**
          Sample a batch of user-item pairs from $\mathcal{D}$
          Update $\tilde{\mathbf{P}}$: $\alpha \leftarrow \alpha - \eta \nabla_\alpha \mathcal{L}_p(\alpha; \theta, \beta \mid \hat{\mathbf{P}})$
      **end for**
      **for** number of steps training the imputation model **do**
          Sample a batch of user-item pairs from $\mathcal{O}$
          Update $\hat{\mathbf{E}}$: $\beta \leftarrow \beta - \eta \nabla_\beta \mathcal{L}_e(\beta; \alpha, \theta \mid \tilde{\mathbf{P}})$
      **end for**
      **for** number of steps training the prediction model **do**
          Sample a batch of user-item pairs from $\mathcal{D}$
          Update $\hat{\mathbf{R}}$: $\theta \leftarrow \theta - \eta \nabla_\theta \mathcal{L}_r(\theta; \alpha, \beta \mid \tilde{\mathbf{P}})$
      **end for**
  **end while**

---

## E  ADDITIONAL SEMI SYNTHETIC EXPERIMENTS DETAILS

**Data Generation Process**. Following the previous studies (Schnabel et al., 2016; Wang et al., 2019; Guo et al., 2021), the detailed preprocessing for the ML-100K [3] dataset is shown as follows.

(1) Complete the full rating matrix using Matrix Factorization (MF) (Koren et al., 2009). Since the rating matrix completed by MF will have a unrealistic high prediction value for ratings of almost all user-item pair, we adjust the proportion of ratings to match a more realistic rating distribution by first sorting all ratings in ascending order, then set ratings below the $p_1$ quantile to 1, set ratings between $p_1$ quantile and $p_2$ quantile to 2, and so on. The adjusted rating matrix contains $R_{u,i} \in \{1, 2, 3, 4, 5\}$ with proportion $[p_1, p_2, p_3, p_4, p_5]$, respectively (Schnabel et al., 2016; Guo et al., 2021).

(2) Set a propensity $p_{u,i} \in (0, 1)$ for each user-item pair with $p_{u,i} = p\alpha^{\min(4, 6 - R_{u,i})}$. In our experiment, $p = 1$ and $\alpha = 0.5$ (Wang et al., 2019; Guo et al., 2021). Then we obtain the ground truth propensity matrix $\mathbf{P}$.

(3) Transfer the adjusted rating matrix to the probability matrix by replacing $R_{u,i} \in \{1, 2, 3, 4, 5\}$ with $r_{u,i} \in \{0.1, 0.3, 0.5, 0.7, 0.9\}$ correspondingly. Because only binary click indicators can be observed, we sample click indicators according to the following Bernoulli distribution:

$$o_{u,i} \sim \mathrm{Bern}(p_{u,i}), \forall (u, i) \in \mathcal{D},$$

where $\mathrm{Bern}(\cdot)$ denotes the Bernoulli distribution. Then we obtain a fully observed observation matrix $\mathbf{O}$ and a ground truth probability matrix $\mathbf{R}$.

**Additional Experiment Results.**

Table 6 shows the results when the inductive bias $b_{u,i}$ is sampled from a uniform distribution $U(0, \alpha)$ with $\alpha = 0.2, 0.3$ and $0.4$ for each user-item pair. Remarkably, UDR, IDR and UIDR also significantly outperform other estimators, which further shows the effectiveness of the proposed estimators.

---

[3]https://grouplens.org/datasets/movielens/100k/

Table 6: Relative errors on ML-100K dataset. For each user-item pair, the inductive bias $b_{u,i}$ is independently sampled from the uniform distribution $U(0, \alpha)$.

| $\alpha = 0.2$ | Naive | EIB | IPS | SNIPS | DR | UDR | IDR | UIDR |
|---|---|---|---|---|---|---|---|---|
| ONE | 0.068 ± 0.003 | 0.112 ± 0.069 | 0.034 ± 0.004 | 0.035 ± 0.004 | 0.024 ± 0.016 | **0.005 ± 0.004*** | **0.013 ± 0.014** | **0.006 ± 0.004*** |
| THREE | 0.078 ± 0.003 | 0.103 ± 0.061 | 0.038 ± 0.003 | 0.039 ± 0.003 | 0.022 ± 0.015 | **0.006 ± 0.005*** | 0.018 ± 0.016 | **0.002 ± 0.001*** |
| FIVE | 0.101 ± 0.004 | 0.124 ± 0.040 | 0.050 ± 0.004 | 0.050 ± 0.004 | 0.028 ± 0.010 | **0.006 ± 0.003*** | 0.014 ± 0.025 | **0.006 ± 0.003*** |
| ROTATE | 0.137 ± 0.001 | 0.019 ± 0.011 | 0.069 ± 0.003 | 0.068 ± 0.002 | 0.004 ± 0.003 | **0.002 ± 0.002** | **0.002 ± 0.001*** | **0.001 ± 0.001*** |
| SKEW | 0.026 ± 0.002 | 0.047 ± 0.035 | 0.012 ± 0.003 | 0.013 ± 0.002 | 0.013 ± 0.009 | **0.004 ± 0.003*** | 0.007 ± 0.007 | **0.003 ± 0.003*** |
| CRS | 0.106 ± 0.002 | 0.106 ± 0.065 | 0.053 ± 0.004 | 0.053 ± 0.003 | **0.012 ± 0.009** | **0.004 ± 0.003*** | 0.012 ± 0.013 | **0.004 ± 0.004*** |

| $\alpha = 0.3$ | Naive | EIB | IPS | SNIPS | DR | UDR | IDR | UIDR |
|---|---|---|---|---|---|---|---|---|
| ONE | 0.067 ± 0.002 | 0.145 ± 0.093 | 0.034 ± 0.003 | 0.033 ± 0.003 | 0.031 ± 0.002 | **0.005 ± 0.003*** | **0.019 ± 0.016*** | **0.006 ± 0.005*** |
| THREE | 0.078 ± 0.003 | 0.207 ± 0.103 | 0.040 ± 0.003 | 0.039 ± 0.004 | 0.044 ± 0.021 | **0.006 ± 0.003*** | **0.014 ± 0.011*** | **0.006 ± 0.005*** |
| FIVE | 0.101 ± 0.003 | 0.227 ± 0.086 | 0.051 ± 0.003 | 0.050 ± 0.003 | 0.050 ± 0.019 | **0.005 ± 0.005*** | **0.017 ± 0.013*** | **0.005 ± 0.003*** |
| ROTATE | 0.137 ± 0.002 | 0.267 ± 0.016 | 0.068 ± 0.002 | 0.069 ± 0.002 | 0.006 ± 0.004 | **0.002 ± 0.002*** | **0.002 ± 0.001*** | **0.002 ± 0.001*** |
| SKEW | 0.025 ± 0.003 | 0.063 ± 0.040 | 0.013 ± 0.004 | 0.012 ± 0.003 | 0.016 ± 0.010 | **0.003 ± 0.002*** | **0.007 ± 0.005*** | **0.004 ± 0.002*** |
| CRS | 0.105 ± 0.002 | 0.144 ± 0.098 | 0.052 ± 0.003 | 0.052 ± 0.003 | **0.016 ± 0.011** | **0.004 ± 0.003*** | 0.019 ± 0.021 | **0.004 ± 0.003*** |

| $\alpha = 0.4$ | Naive | EIB | IPS | SNIPS | DR | UDR | IDR | UIDR |
|---|---|---|---|---|---|---|---|---|
| ONE | 0.069 ± 0.002 | 0.207 ± 0.122 | 0.036 ± 0.003 | 0.035 ± 0.026 | 0.045 ± 0.003 | **0.006 ± 0.005*** | **0.006 ± 0.027*** | **0.004 ± 0.004*** |
| THREE | 0.079 ± 0.002 | 0.257 ± 0.137 | 0.040 ± 0.003 | 0.040 ± 0.003 | 0.055 ± 0.028 | **0.006 ± 0.005*** | **0.017 ± 0.014*** | **0.003 ± 0.002*** |
| FIVE | 0.102 ± 0.003 | 0.253 ± 0.133 | 0.052 ± 0.003 | 0.052 ± 0.003 | 0.058 ± 0.030 | **0.010 ± 0.009*** | **0.017 ± 0.025*** | **0.006 ± 0.004*** |
| ROTATE | 0.137 ± 0.001 | 0.039 ± 0.017 | 0.068 ± 0.002 | 0.068 ± 0.002 | 0.008 ± 0.004 | **0.002 ± 0.001*** | **0.003 ± 0.003*** | **0.002 ± 0.001*** |
| SKEW | 0.026 ± 0.002 | 0.115 ± 0.058 | 0.014 ± 0.003 | 0.013 ± 0.003 | 0.016 ± 0.003 | **0.007 ± 0.007*** | **0.011 ± 0.011*** | **0.005 ± 0.004*** |
| CRS | 0.108 ± 0.002 | 0.176 ± 0.120 | 0.056 ± 0.002 | 0.055 ± 0.002 | 0.022 ± 0.012 | **0.009 ± 0.007*** | **0.012 ± 0.011*** | **0.004 ± 0.003*** |

Note: * means statistically significant results (p-value $\leq 0.05$) using the paired-t-test compared with the best baseline. We bold the best three results and underline the best baseline result.

## F  DATASET PREPROCESSING, EXPERIMENTAL PROTOCOLS AND DETAILS.

**Dataset and Preprocessing.** Following the previous studies (Wang et al., 2019; 2021; Chen et al., 2021), we conduct extensive experiments on three real-world datasets: COAT[4], MUSIC[5], and KUAIREC[6] (Gao et al., 2022). COAT dataset contains 290 users and 300 items with 6,960 biased ratings and 4,640 unbiased ratings. MUSIC dataset contains 15,400 users and 1,000 items with 311,704 biased ratings and 54,000 unbiased ratings. COAT and MUSIC are both five-scale datasets, and we binarize ratings less than three to 0, otherwise to 1. KUAIREC is a large-scale fully exposed industrial dataset collected from a short video sharing platform, which contains 4,676,570 video watching ratios from 1,411 users to 3,327 videos. For KUAIREC dataset, we binarize the video watching ratios less than one to 0, otherwise to 1.

**Experimental Protocols and Details.** We use three widely adopted evaluation metric, AUC, NDCG@K and F1@K, where K is set to 5 for COAT and MUSIC and 50 for KUAIREC. All the experiments are implemented on PyTorch with Adam as the optimizer. For all experiments, we use GeForce RTX 3090 as the computing resource. Logistic regression is used as the propensity model for all the methods with propensity. We tune the learning rate in $\{0.001, 0.005, 0.01, 0.05, 0.1\}$ and the batch size in $\{32, 64, 128, 256\}$ for COAT and $\{1024, 2048, 4096, 8192\}$ for MUSIC and KUAIREC. We tune the embedding dimension in $\{2, 4, 8, 16, 32, 64\}$ for COAT and $\{16, 32, 64, 128, 256, 512\}$ for MUSIC and KUAIREC. Moreover, we tune the hyper-parameter $\gamma$ in $\{1e-6, 5e-5, 1e-5, ..., 1e-1\}$.

---

[4]https://www.cs.cornell.edu/~schnabts/mnar/

[5]http://webscope.sandbox.yahoo.com/

[6]https://github.com/chongminggao/KuaiRec

