# OpenReview forum: "Towards Relaxing the Unbiasedness Condition of Doubly Robust Estimators for Debiased Recommendation"
_ICLR.cc/2024/Conference — ICLR 2024 Conference Withdrawn Submission_

### Official Review · Reviewer_BXqD · 2023-10-24

**Soundness:** 2 fair
**Presentation:** 3 good
**Contribution:** 2 fair
**Rating:** 3
**Confidence:** 4

**Summary:**

The paper introduces a novel doubly robust estimator to address observational bias in collaborative filtering.\
It focuses on the effectiveness of the method, particularly in handling inaccurate pseudo-labels. \
The authors propose debiasing techniques with constraints on propensity scores, aiming to ensure unbiasedness when the pseudo labels deviate from the true labels with an arbitrary inductive bias. \
The paper combines theoretical analysis and experimental evaluations to demonstrate the method's effectiveness in comparison to benchmark approaches.

**Strengths:**

1. Comprehensive Background: The authors provide a well-explained introduction and extensive coverage of prior related work, enhancing the paper's accessibility to readers by providing context and a clear understanding of the research landscape.

2. Novel and Effective Method: The proposed method effectively addresses the issue of pseudo-labels deviating from true values with specific inductive bias.

3. Theoretical and Empirical Validation: The paper combines theoretical analysis with empirical results, covering both semi-synthetic and real-world datasets, demonstrating the method's technical soundness and its superior performance compared to existing approaches.

**Weaknesses:**

1. Limited Motivation: The paper may lack novelty in the central idea. Is there any empirical evidence that existing pseudo-labeling suffers from user/item specific inductive bias? If users have an inductive bias in the training set, the user should have the exact inductive bias in the test set.

2. Strong Assumptions: The assumptions regarding inductive biases for users and items might not be realistic. The paper assumes constant inductive biases for users across all items, which may not hold true in real-life scenarios. Users typically exhibit varying preferences for different items.

3. Unclear Propensity Score Calculation: The paper lacks a clear description of how propensity scores were calculated. There is a need for a more detailed explanation.

**Questions:**

1. Personally, I cannot understand the intuitive concept of the experiment with the synthetic dataset. Can you explain the intuition?
2. Is there any empirical or theoretical evidence for the motivating inductive bias problem?

---

### Official Review · Reviewer_aJVp · 2023-10-30

**Soundness:** 3 good
**Presentation:** 3 good
**Contribution:** 2 fair
**Rating:** 5
**Confidence:** 4

**Summary:**

The paper presents a new doubly robust estimator for the missing-not-at-random bias.
The authors propose constraints on propensity scores to handle pseudo-labels deviating from the true value with the user-specific bias.
They provide theoretical analysis and practical experiments, showing how their approach outperforms benchmark methods.

**Strengths:**

- the solution (a constrained propensity model) is simple yet effective.
- the authors provide theoretical analyses on the unbiasedness and the variance of the proposed estimator.

**Weaknesses:**

- the target problem is too specific and minor. The authors noted that UIDR can effectively alleviate the "inaccurate pseudo-labeling problem" in the previous DR estimators. However, they only treat the situation where the pseudo-labelings deviate from the true labels with arbitrary user-specific inductive bias. This assumption looks quite unrealistic as the user-specific inductive bias is assumed to be equivalent for every item.
- the proposed procedure is not well-motivated. The authors put constraints on the propensity model, not the imputation model, in order to tackle the inaccurate imputation model. If the imputation model is inaccurate the straightforward remedy would be either adjusting the imputation model itself or designing a new loss function robust to the inaccurate imputed errors. If we adjust the propensity model for the imputation model, the accuracy of the propensity model can be harmed.

**Questions:**

- please refer to weaknesses.
- ex) If we adjust the propensity model for the imputation model, how does the accuracy of the propensity model become?

---

### Official Review · Reviewer_uhbY · 2023-11-01

**Soundness:** 3 good
**Presentation:** 3 good
**Contribution:** 3 good
**Rating:** 6
**Confidence:** 3

**Summary:**

The paper presents a study on the challenges and potential solutions associated with debiasing the recommender system models due to the sampling selection bias in the process of data collection. The authors propose several novel doubly robust estimators that are unbiased.
These estimators are unbiased for arbitrary user-specific, item-specific inductive bias, and even both. Authors also theoretically prove these estimators’ unbiasedness. Besides, they propose a propensity reconstruction learning approach that adaptively updates the constraint weights to meet the constraints of the proposed UIDR estimator.

**Strengths:**

(1) The authors introduce a series of innovative double robustness (DR) estimators through a rigorous theoretical framework. These estimators maintain their unbiased nature even when pseudo labelings diverge from the true labels, accommodating arbitrary and unknown biases specific to users, items, or a combination thereof. This represents a significant stride in addressing user-specific and item-specific inductive biases, showcasing the adaptability and robustness of the proposed methods.

(2) In a further extension of their work, the authors present a principled propensity reconstruction learning strategy, which adeptly utilizes an attention mechanism to adaptively update the constraint weights. This approach not only enhances the adaptability of the model but also ensures that the variance of the DR estimators remains within controllable and manageable bounds. This aspect of the work underscores the authors’ commitment to developing robust and reliable estimators, contributing to the stability and efficacy of the proposed methods.

(3) The paper’s empirical validation is robust, encompassing semi-synthetic experiments that attest to the effectiveness of the proposed methods in scenarios involving arbitrary user-specific and item-specific inductive biases. This is a notable achievement, as previous methodologies have fallen short in providing unbiased estimates of the ideal loss under these conditions. Additionally, the authors extend their validation to real-world contexts, conducting comprehensive experiments across three real-world datasets. These experiments serve to highlight the tangible advantages and superior performance of the proposed methods, solidifying the paper’s contributions to the field.

**Weaknesses:**

(1) While the theoretical foundation appears robust, there is a potential concern regarding the complexity and practicality of implementing such estimators in real-world scenarios when we have large U and large I. The paper could benefit from a more detailed discussion on the potential challenges and limitations associated with these novel DR estimators, providing a more balanced and critical perspective.

(2) The introduction of a principled propensity reconstruction learning approach, utilizing an attention mechanism to adaptively update constraint weights, is indeed a novel contribution. However, the claim that the variance of the DR estimators is highly controllable and manageable warrants a more rigorous scrutiny. Can we see any tradeoff between the bias and variance, since sometimes we want to minimize the MSE in the ML community when you mentioned controllable and manageable.

**Questions:**

1. In section 2, it seems that $r_{u,i}$ in [0,1] rather in R, which is the rating/

2. After corollary 1, “the biases of the proposed DR estimators are still strictly smaller than the previous DR” , Is this toy example realistic? Could you please show a real example here?

3. For the optimization problem, do we have a constraint that $\tilde{p}$ < 1?

4. What is A? What is b? What is $s_{u}$ and $t_{I}$? How do we learn it? Why $s_{u}$ has been applied tanh but $t_i$ is not been applied?

---

### Official Review · Reviewer_d1UZ · 2023-11-05

**Soundness:** 3 good
**Presentation:** 3 good
**Contribution:** 3 good
**Rating:** 8
**Confidence:** 3

**Summary:**

This paper proposes User-DR, Iten-DR, and User-Item-DR for unbiased recommendation. The proposed methods have both strong theoretical guarantees and improvement in comparison with the baselines.

**Strengths:**

1. Strong theoretical guarantee on the proposed method
2. The experimental studies are solid. I would like to point out that a ~10% improvement in NDCG@5 is very significant for unbiased recommendation. Note that the machine learning model is not changed in this paper. To be more accurate, only the debiased method is different from the baselines if I understand correctly!

**Weaknesses:**

1. Some baselines are lacking introduction. For example, the best baseline, ESCM-DR, is not introduced in detail.
2. The running time (or time complexity) is lacking in analysis in the paper. I think this paper can be improved if ESCM-DR is slower than the proposed algorithms.
3. The evaluation matrics, AUC, NDCG, and F1, are missing clear definitions (can be in the appendix).

**Questions:**

1. Can the authors provide some detailed explanations for ECSM-DR and Multi-DR?
2. Any record of the running time (or time complexity) can be provided?
3. Can the authors provide a rigid definition for F1 measure?
4. Can the authors also provide the experimental results similar to Table 3 for NDCG and F1 @ other values in the appendix? At least for Coar and music datasets, Top-1 should be also an important measure.